# Influence of Foundation Quality on the Stress in the Elements of Steel Façade Scaffolding

**Ewa Błazik-Borowa** , **Paulina Jamińska-Gadomska** and **Michał Pieńko** *

Department of Structural Mechanics, Faculty of Civil Engineering and Architecture, Lublin University of Technology, 20-618 Lublin, Poland; e.blazik@pollub.pl (E.B.-B.); p.jaminska@pollub.pl (P.J.-G.)
* Correspondence: m.pienko@pollub.pl; Tel.: +48-81-538-44-37

**Abstract:** The purpose of this paper is to show the influence of incorrect scaffolding foundations on the stress in their elements. Static stress analysis was performed for exemplary steel façade scaffolding. The scaffolding was formed using the Plettac 70 system and was composed of 16 modules and 13 working levels. The total dimensions of the scaffolding were $45.0 \times 26.36 \times 0.74$ m. The scaffolding was set up partly on concrete and partly on a created ground classified as coarse sand with discontinuous graining. The boundary conditions modelling the foundation considered the heterogeneity of the ground both along the scaffolding and in the direction perpendicular to the façade. The effect of uneven subsidence on the scaffolding frames was checked, adopting a constant stiffness of 3500 kN/m in half of the supports, while in the rest of them the stiffness varied from 35 to 3500 kN/m. Due to additional bending moments, normal stresses in stands and transoms of the frames increased. Incorrect scaffolding foundation has the greatest negative effect on normal forces in anchors and bracings. Because these elements are responsible for the stability of the scaffolding, their damage may result in scaffolding failure and would certainly lead to a reduction of the values of free vibration frequencies, thus resulting in the discomfort of the workers on the scaffolding and a lack of safety.

**Keywords:** scaffolding; subsoil; foundation; static analysis; structural safety

---

## 1. Introduction

The majority of construction sites use scaffolding for work at a certain height and in places difficult to access. Therefore, even though scaffolding is a temporary structure, it is one of the most important parts of the work safety organization at construction sites. Due to a high risk of failures and incidents, its design and assembly should be carefully prepared, which is why unusual scaffolding, i.e., high or with complex geometry, needs specific design solutions [1–3] focused on the points of support and anchoring, which have a direct impact on the level of stress of individual scaffolding elements and the risk of scaffolding disaster. Analyses regarding the impact of the anchoring distribution and the nature of conducted work can be found in [4]. Forces in anchors are strictly dependent on the geometry [5] and height of the scaffolding, objects occurring in its immediate vicinity [6], as well as on safety nets and claddings [7] attached to the scaffolding, because the main impact on the forces in anchors is caused by the wind [8]. Another aspect affecting the anchoring stress in inaccuracies in the structure's geometry (imperfections) [9]. The scaffolding foundation also has influence on the structural stability and stress of individual elements. Incorrect scaffolding foundation on wet ground without drainage was indicated as one of the main causes of the 2001 disaster [10] in which two people were killed and seven were injured. However, there is a lack of research in the literature containing information about scaffolding, foundations which clearly have significant impact on the stress in scaffolding elements and safety of the whole structure.

In the years 2016 to 2018, the project entitled: "Modelling of risk assessment of construction disasters, accidents, and dangerous incidents at workplaces using scaffoldings" (ORKWIZ) was conducted. The main aims of the research were to perform scaffolding tests in the aspects of ensuring employee safety [11–21], studies on the safety culture of employees [22,23], an analysis of the causes of over 180 accidents on scaffolding [24,25], and research on the amount of scaffolding used in Poland depending on the season [26]. Additionally, examples of the errors of scaffolding assembly that negatively affect their carrying capacity were shown in paper [16]. These errors included improper scaffolding setting on ground, geometric imperfections as a result of careless assembly, incorrect anchor arrangement, improper connection of anchors with scaffolding, and damage to elements. The issue of correct scaffolding foundation is practically completely marginalized, which can be proven by the fact that only for one of the tested scaffolding structures was a soil test was carried out before its erection.

Scaffolding accidents and potentially hazardous situations can occur because of the condition of the structure. It is obvious that the poor technical condition of the handrails or platforms threatens the safety of users, but also the preparation of the ground on which the scaffolding is placed is particularly important for safety. References [12] and [16] show the impact of incorrect foundation and installation accuracy on work comfort and risk of accidents. These works describe the cause and effect relationships between the condition of the ground and the functionality of the scaffolding. If the scaffolding is placed on a ground without uniform physical properties, or the ground does not have an adequate load-bearing capacity, then the scaffolding can become unevenly settled and suspended on anchors. An increase in geometric imperfections in scaffolding and loosening of the anchors can be observed. As a consequence, it causes a reduction in scaffolding stability and safety. The purpose of this article is to show in detail one of the listed problems, namely the direct impact of the quality of ground on the normal stress in the elements of the exemplary scaffolding. This paper is the first work in which a qualitative and quantitative analysis of this issue is made. Usually, on construction sites in Poland, a primitive and subjective technique for checking ground stability is used. The fitter stands on the ground on which the scaffolding is to be placed, and if the ground does not collapse, then the ground is assessed as suitable for the scaffolding. This can be performed with low, uncomplicated scaffolding. However, with the current trend of building higher scaffolding with more and more complicated geometry, this approach is insufficient and, as we show in this paper, it can lead to scaffolding failure.

## 2. Description of Research Methods

### 2.1. Description of the Scaffolding at the Construction Site

A total number of 120 scaffolding structures was taken into account in the ORKWIZ project. The tested scaffoldings were used for construction works all over Poland. The structures had various dimensions and were set on different foundations. A common feature was that they comprised frame, one plane scaffolding. However, in this paper the impact of scaffolding foundation quality on the scaffolding stress is shown for the example of one scaffolding structure, named L12 (Figure 1). This scaffolding was chosen because during the tests it was found that part of the mudsills was pressed into the ground, and some of them were located on concrete and some on uncompacted ground (Figure 2). On the one hand, pressed mudsills meant that the load capacity of the ground was exceeded, and on the other hand, two types of foundations meant that the ground did not guarantee uniformity of parameters. In addition, the scaffolding is long enough to enable research of the impact of settlement on selected parts of the structure, the results can be treated as representative for other frame scaffolding structures.

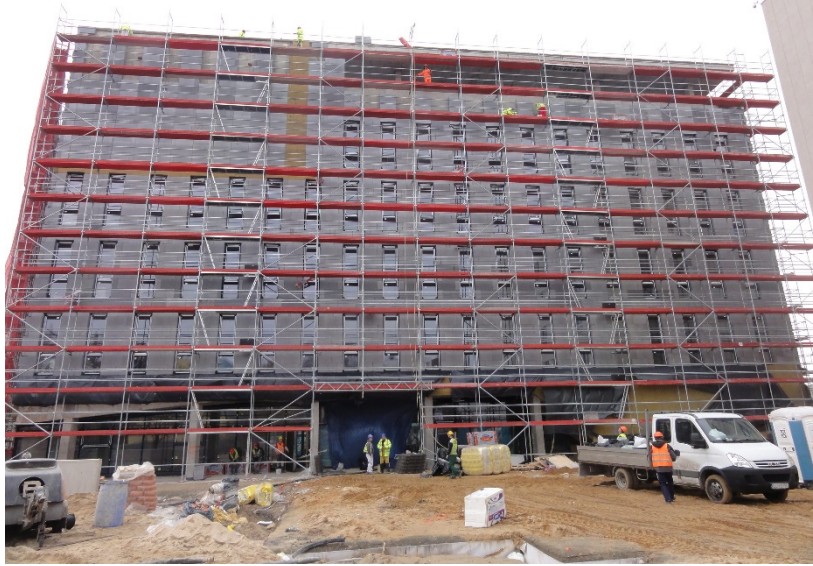

**Figure 1.** The L12 scaffolding.

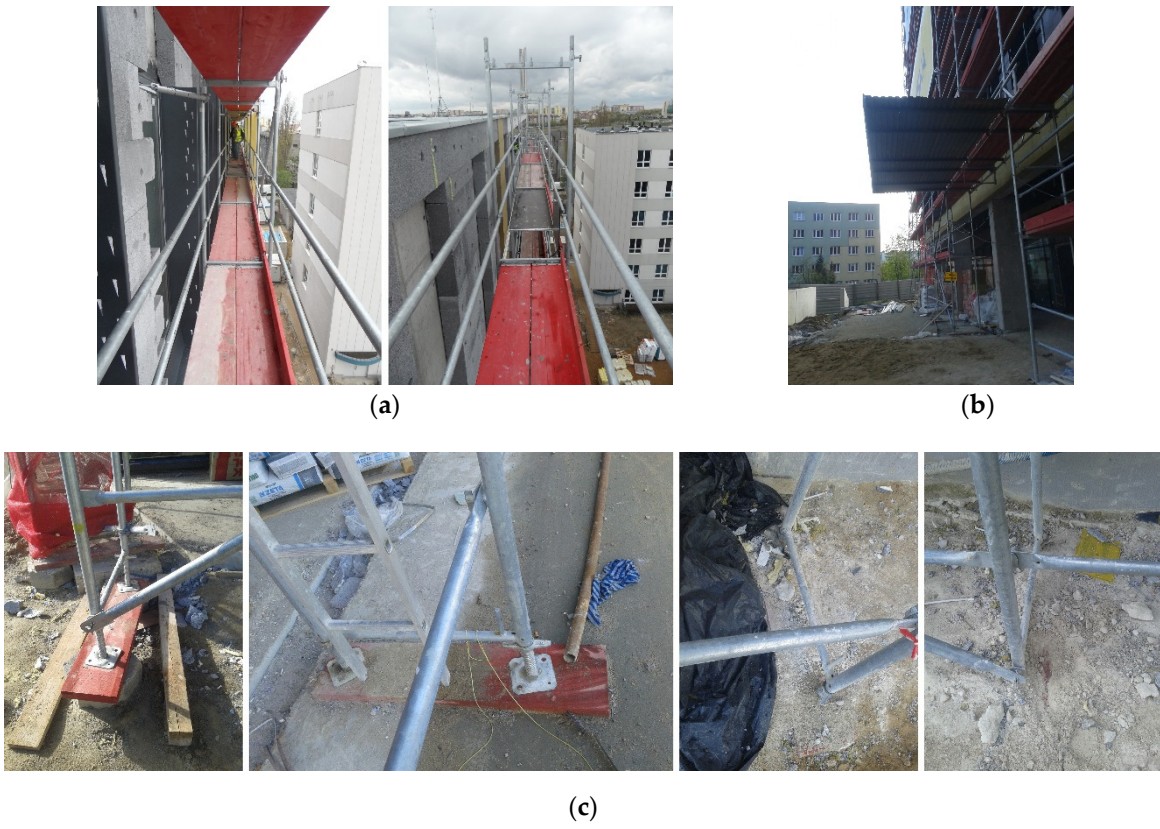

**Figure 2.** Elements of L12 scaffolding: decks and anchors (**a**); protective roof (**b**); mudsills (only the second frame is set correctly) (**c**).

An inventory of scaffolding along with other tests such as measurements of forces in stands, pull-out forces in anchors, scaffolding loads (service load, wind action, dynamic interactions, e.g., traffic), scaffolding point acceleration, and bearing capacity of ground was made in April 2017.

The steel façade scaffolding was formed using the Plettac 70 system (ALTRAD plettac assco GmbH, Plettenberg, Germany) composed of 16 modules and 13 working levels. The total width, length and height of scaffolding were 0.74, 45.0, and 26.36 m, respectively. The $f_{yk}$, the characteristic yield strength

of steel, from which the scaffolding components were made, was equal to 320 MPa for stands and 235 MPa for other elements. A photo of the scaffolding on the construction site is shown in Figure 1, while Figure 2 presents the most important structural elements of the scaffolding including mudsills.

A geodetic inventory of the scaffoldings was carried out during the research. The measurement methodology has been described in detail in [18]. The direct results of the geodetic measurements were locations of points on the surface of the stands. Two points at one height or two points at two heights were measured for each stand. On this basis, the centers of the circles forming the cross-section perimeter at a given height, the axes of the scaffolding stands, and finally the locations of their connections were determined. Geodetic surveys show that the scaffolding geometry was significantly different from the ideal layout. The distribution of imperfections on the outer plane of the scaffolding is shown in Figure 3. The average imperfection value, defined as the distance between the actual position of the connection of the stands in two frames and its position in ideal geometry, was 8.8 cm. The maximum imperfection value, which was determined from all imperfections measured on the L12 scaffolding was equal to 18.3 cm. It was obtained for the top node of the frame, at the outside plane of the scaffolding. This frame was located at the at the highest level, at the end of the right side of the scaffolding. The maximum imperfection was caused by node translation along the scaffolding by 6.5 cm; scaffolding moving away from the wall by 12.7 cm; and relative settlement of the scaffolding segments, which caused 11.4 cm difference of the levels in the extreme decks.

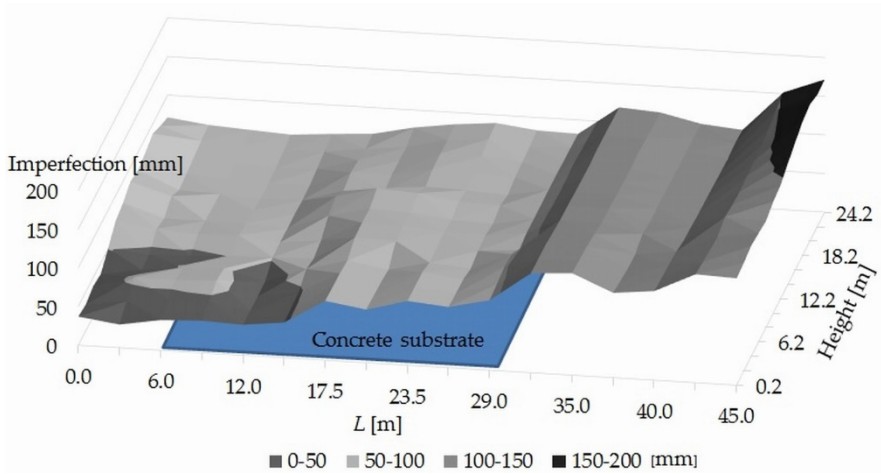

**Figure 3.** Distribution of imperfections on the outer plane of the L12 scaffolding.

The second group of significant tests carried out on scaffolding were scaffolding point acceleration measurements performed during its free vibrations. Measurement points were located under the highest deck level. The excitation was introduced at the level of decks below the measuring points. Fast Fourier Transforms, i.e., signal strengths, were determined based on the time series of accelerations depending on the vibration frequency. From the graphs obtained this way, the free vibration frequency values were read, which correspond to the ordinates of points with signal power peaks. The research methodology and the process of determining the frequency of free vibrations was described in [15,19,27]. For the L12 scaffolding the first free vibration frequency was equal to 1.94 Hz. The first form corresponded to the vibrations along the scaffolding. High imperfection values and low frequency of free vibrations are, among others, the result of incorrect scaffolding foundation. As can be seen in Figure 3, the shifts in nodes are similar for modules erected on concrete, whereas outside this area, there are clear differences between subsidence of frames. The thesis that low frequencies are also the result of incorrect foundation is based on the results of the process of selecting the boundary conditions of the scaffolding. This methodology is based on comparing the natural frequencies with those obtained from the measurements. In the case of L12 scaffolding, it was found that obtaining such low natural frequencies in FEM (Finite Element Method) analyses is possible only if the anchors' connection to the

wall is modelled as hinged support. The methodology used to select the boundary conditions in the scaffolding was described in [27].

The dynamic deformation modulus of subsoil $E_{vd}$ was measured using the lightweight deflectometer ZORN ZFG3000 of ZORN Instruments GmbH & Co. KG from Stendal, Germany. Measurements were made on uncompacted ground, at each frame in between its stands. Based on the visual assessment, the subsoil was found to be a created ground consisting of coarse sand with discontinuous graining. For this type of subsoil, the stiffness of supports was determined from the formula:

$$k_Z = \frac{E_2 A_p}{1\text{m}} \tag{1}$$

where $A_p$ is equal to half of the mudsill area and $E_2$ is the secondary deformation modulus (cf. [28]) equal to

$$E_2 = 2.19 E_{vd} - 5.07 \text{ MPa} \tag{2}$$

Figure 4 shows the results of the $E_{vd}$ module and the resultant stiffness of the support $k_Z$, which were used in further calculations. Values of the $E_{vd}$ module are given on the left axis, whereas values of stiffness $k_Z$ are given on the right axis. The lack of data on soil parameters between 5 and 30 m results from the fact that from the 3rd to 11th stand, scaffolding was placed on a concrete base (marked on the graph as a "concrete substrate"). The design's average load capacity of uncompacted ground was determined in accordance with EN 1997-1 [29], $q_{Rd}$ = 161.7 kN/m$^2$. This value was determined for the average module $E_{vd}$ = 14.5 MN/m$^2$ using the procedure described in [15] and [30].

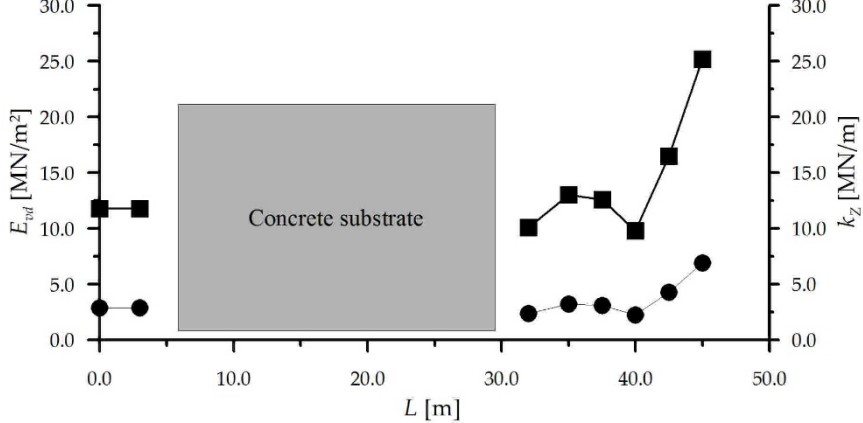

**Figure 4.** Test results from the lightweight deflectometer ZORN ZFG3000: ■ $E_{vd}$—dynamic deformation modulus of subsoil, ● $k_Z$ —stiffness of support, $L$—distance measured along the scaffolding.

*2.2. Numerical Analyses*

Numerical analyses were conducted for three configurations of the L12 scaffolding:

- C1—real geometry and boundary conditions modelling the ground in accordance with the test results, i.e., if the mudsills were located on compacted ground, hinge supports were adopted, and if not, supports blocking movements in two horizontal directions and a spring support for a vertical direction with stiffness $k_Z$ (shown in Figure 5) were used.
- C2—ideal geometry and foundation model, consisting of supports blocking horizontal movement and spring supports in the vertical direction with different values of stiffness (Figure 4).
- C3—ideal geometry and foundation model, consisting of hinge supports blocking movements in three directions.

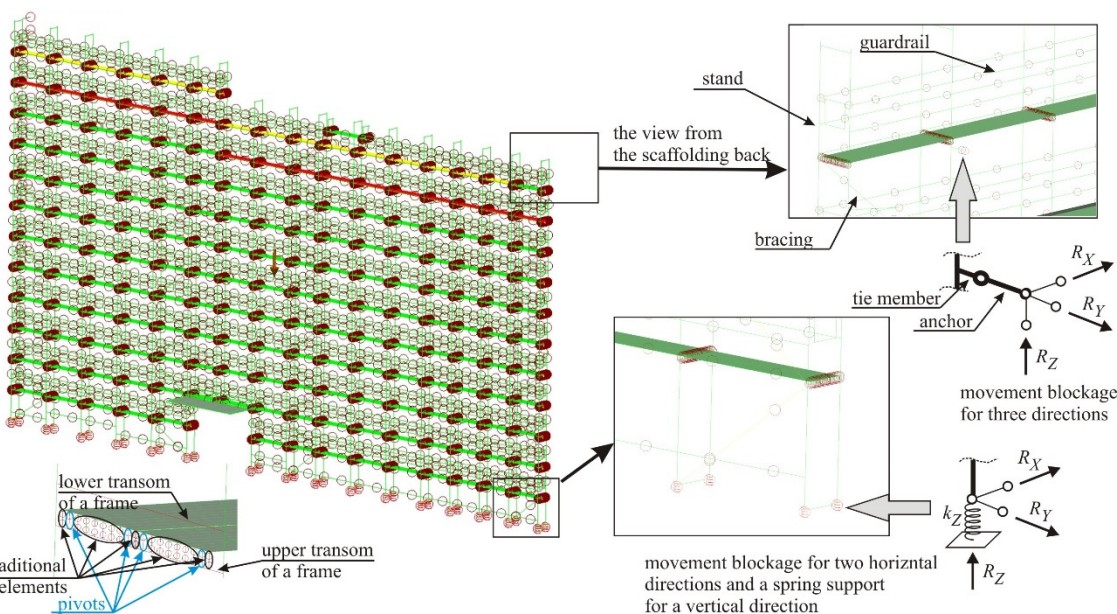

**Figure 5.** Static scheme of L12 scaffolding, configuration C2.

Figure 5 presents the static scheme of the scaffolding for the configuration C2. The first configuration, C1, was used to verify the model and to show the actual state of the scaffolding caused by incorrect foundation. The configurations C2 and C3 were used to study the impact of ground quality on scaffolding stress and development of general conclusions.

The analyses were performed for two cases regarding uneven parameters of the ground. The first one concerned uneven settlement perpendicular to the building. Scaffolding is usually set on the backfilled subsoil, which is near the walls, i.e., at the location of the internal stands, and is often uncompacted but shows much better parameters at the location of the external stands. The correct use of mudsills eliminates such irregularities, but unfortunately, as shown for example in references [13] and [16], in practice mudsills are not always placed under both stands of frame in a direction perpendicular to the building. The effect of uneven subsidence on the scaffolding frames in a direction perpendicular to the facade was checked based on the assumption that the rigidity of the elastic supports on the stands external to the façade was constant and equal to $k_Z$ = 3500 kN/m, while on the internal ones it varied from 35 to 3500 kN/m.

The second case concerned the impact of uneven settlement of frames along the façade. As presented in Figure 4, different modules of elasticity of the foundation were obtained under almost every stand. In this case, in every second pair of supports (under every other frame) the stiffness of supports was assumed equal to $k_Z$ = 3500 kN/m, and in the remaining ones it varied from 35 to 3500 kN/m.

Other elements of the numerical model are the same for all configurations. As already mentioned in the previous paragraph, the anchors in the wall were modelled as hinged supports. The following FEM elements were used in the model:

- beam elements modelling base jacks, stands, transoms, handrails, anchor connectors, anchors, bracing, bolts, and elements of steel girders located above the gate,
- truss elements with high stiffness and zero mass density to model deck support on transoms,
- plate elements modelling wooden decks.

For wooden decks the following data were adopted: mass density 0.475 t/m$^3$, thickness 0.044 m, Young's modulus 9·10$^6$ kPa, Poisson's ratio 0.2. The beam elements modelled steel components with the following parameters: mass density 8.0 t/m$^3$, Young's modulus 2.05·10$^8$ kPa, Poisson's ratio 0.3. The mass density of the elements was increased to take into account the total weight of the other

elements of the scaffolding, like joints, gusset plates, toe-boards, etc. The geometric characteristics of cross-sections of beam elements are summarized in Table 1.

**Table 1.** The geometrical characteristics of element cross-sections.

| Component | $A$ (cm$^2$) | $J_1$ (cm$^4$) | $J_2$ (cm$^4$) | $J_3$ (cm$^4$) | $W_2$ (cm$^3$) | $W_3$ (cm$^3$) |
|---|---|---|---|---|---|---|
| Base jack ϕ 38 | 3.2711 | 7.5358 | 3.7679 | 3.7679 | 2.2164 | 2.2164 |
| Standard, ϕ 48.3 × 2.7 | 3.8679 | 20.1775 | 10.0888 | 10.0888 | 4.1775 | 4.1775 |
| Lower transom ϕ 33.7 × 2.6 | 2.5403 | 6.1854 | 3.0927 | 3.0927 | 1.8354 | 1.8354 |
| Upper transom rectangular pipe 50 × 35 × 2 | 3.2400 | 12.3904 | 6.4447 | 11.3132 | 3.6827 | 4.5253 |
| Bracing ϕ 48.3 × 2.6 | 3.7328 | 19.5531 | 9.7765 | 9.7765 | 4.0483 | 4.0483 |
| Guardrail ϕ 38 × 1.8 | 2.0471 | 6.7230 | 3.3615 | 3.3615 | 1.7692 | 1.7692 |
| Tie member ϕ 48.3 × 3.2 | 4.5340 | 23.1713 | 11.5857 | 11.5857 | 4.7974 | 4.7974 |
| Anchorage ϕ 12 | 1.1310 | 0.2036 | 0.1018 | 0.1018 | 0.1696 | 0.1696 |
| Pivot ϕ 12 | 1.1310 | 0.2036 | 0.1018 | 0.1018 | 0.1696 | 0.1696 |
| Girder—horizontal element ϕ 48.3 × 3.2 | 4.5340 | 23.1713 | 11.5857 | 11.5857 | 4.7974 | 4.7974 |
| Girder—lattice ϕ 38 × 2 | 1.8400 | 1.9200 | 1.1100 | 2.1500 | 1.1100 | 1.4300 |

$A$—cross sectional area, $J_1$—torsional resistance, $J_2$ and $J_3$ — moments of inertia, $W_2$ and $W_3$—section modulus.

The connections between the individual components were modelled as follows:

- base jack with base jack—rigid joint,
- stand with stand—rigid joint,
- stands with transoms—rigid joints,
- handrails with stands—pin-connections,
- bracings with stands—pin-connections,
- bolts with transoms—rigid joints,
- bolts with decks—pin-connections,
- girder elements with girder elements—rigid joints,
- girder elements with stands—rigid joints,
- anchor connectors with stands—rigid joints,
- anchor connectors with anchors—pin-connections.

All calculations were made using linear static analyses and Autodesk Simulation Multiphysics 2013 software, which used FEM. The use of linear material models could show the differences in normal stresses for each case and configuration. The use of a non-linear model would cause all calculations to end at the material yield strength, which cannot capture the sensitivity of the calculation results to the unevenness of the settlement. Large displacements were also not included. The critical buckling load analysis of scaffolding showed that the smallest buckling load multiplier equal to 4.06 was obtained for configuration C1, which means that according to EN 12810-2 [31], the first order theory in scaffolding design could be used.

## 3. Results and Discussion

### 3.1. Analysis of the Structure in Real Configuration

The results of the tests conducted at the construction site showed that the scaffolding foundation was incorrect. Subsoil parameters indicated its heterogeneity. In the results, remarkably high imperfections and a low frequency of free vibrations were observed. Further information about behavior of the scaffolding in real conditions at the construction site were obtained based on computer calculations of the scaffolding in the C1 configuration.

Figure 6 shows normal stresses in the structural elements of the scaffolding in the C1 configuration resulting from four load configurations applied in accordance with EN 12811-1 [32] for the 3rd load class and determined from the formula:

$$\sigma = \frac{N}{A} + \frac{M_2}{W_2} + \frac{M_3}{W_3} \tag{3}$$

where $N$ is normal force, $M_2$ and $M_3$ are bending moments about the main axes of the cross-section, $A$ is cross sectional area., and $W_2$ and $W_3$ are section modules at bending about the principal axes.

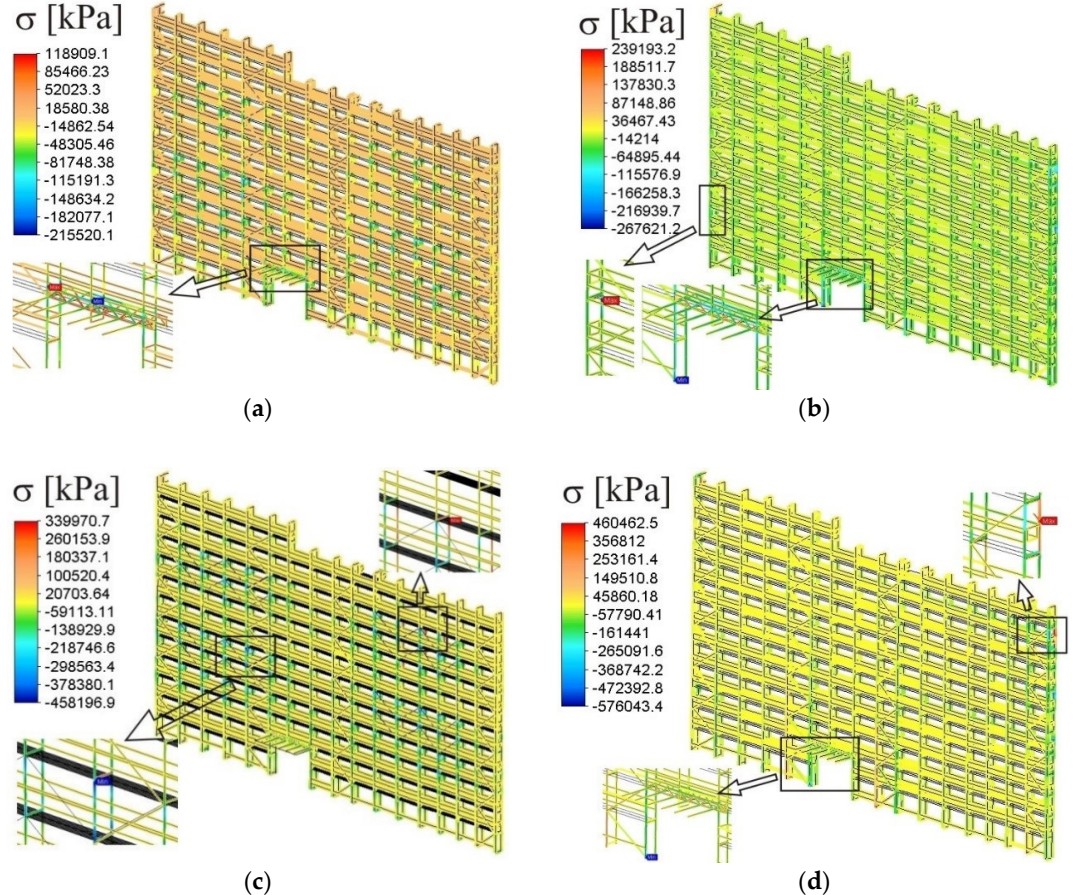

**Figure 6.** Normal stress in L12 scaffolding for the C1 configuration, caused by self-weight and: service load and wind perpendicular to the facade according to [32] (**a**); service load and wind parallel to the facade according to [32] (**b**); perpendicular wind according to [33] (**c**); parallel wind according to [33] (**d**).

As can be seen in Figure 6, normal stresses in configurations with service load (Figure 6a,b) did not exceed the yield point. However, in the cases with the wind as the main load (Figure 6c,d), the yield point was exceeded. Of course, this scaffolding should not be approved for use. The failure did not happen only due to the fact that during the scaffolding operation on the construction site, the wind speed did not reach the Eurocode specified values.

The maximum pressure on one mudsill transmitted through base jacks from two stands of one frame was equal to 58.2 kN. This pressure was obtained for the loads combining scaffolding self-weight and wind action perpendicular to the scaffolding. The resultant ground pressure was equal to 211.7 kN/m², the value greater than the permissible value of $q_{Rd}$ (see Section 2.1). As previously stated, there was no strong wind during the scaffolding operation. However, there were loads large enough for some mudsills to be pressed into the ground, as shown in Figure 2c.

*3.2. Static Analysis of Scaffolding on a Heterogeneous Foundation*

The impact of ground heterogeneity on the stress of scaffolding elements was analyzed on the basis of changes in displacements (Figure 7), normal forces in base jacks, bracings and anchors (Figure 8), normal stresses in stands, transoms, and girders (Figure 9) depending on the value

$$\mu = k_{Z\,min} / k_{Zmax} \tag{4}$$

where dynamic modulus $k_{Zmax}$ = 3500 kN/m and $k_Z$ takes values according to Table 2. The analysis was carried out for configuration C2, taking into account uneven settlement of the ground, while configuration C3, in which all the supports were hinge supports blocking movements in three directions, was used as a reference.

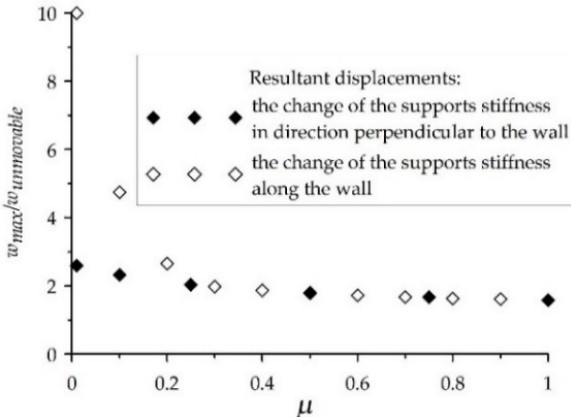

**Figure 7.** Maximum resultant displacements depending on $\mu$ value.

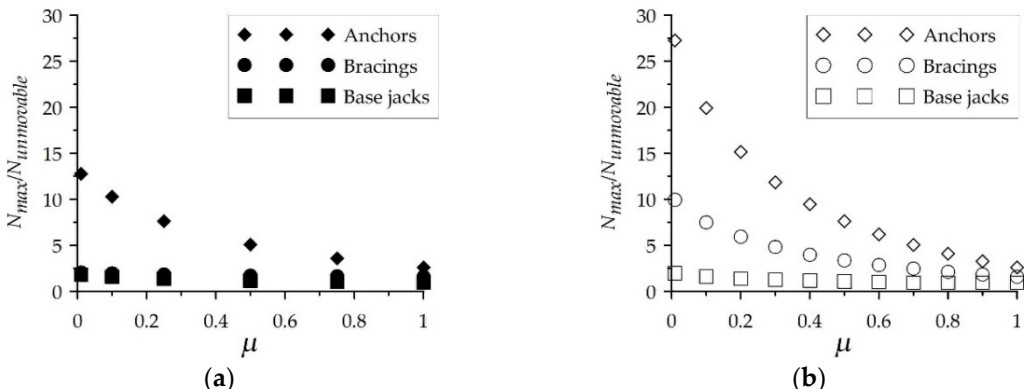

**Figure 8.** Maximum normal forces depending on the $\mu$ value with ground heterogeneity in a direction: perpendicular to the facade (**a**); parallel to the façade (**b**).

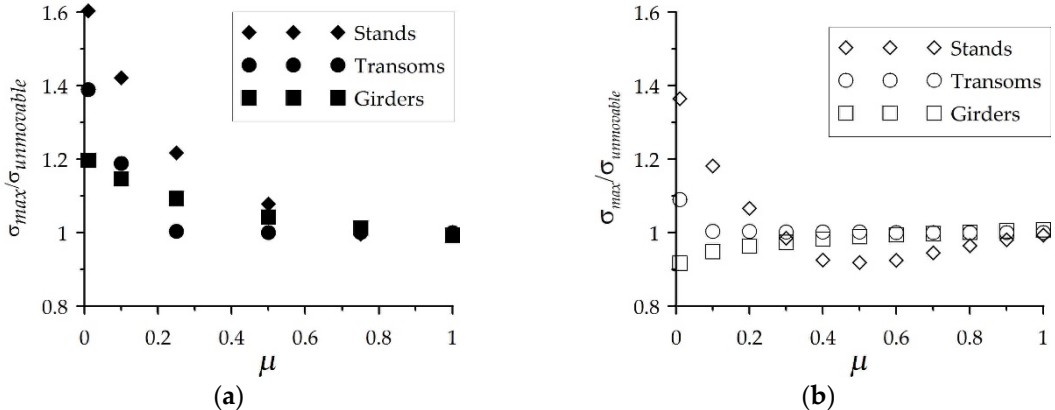

**Figure 9.** Extreme normal stresses depending on the $\mu$ value with ground heterogeneity in a direction: perpendicular to the facade (**a**); parallel to the façade (**b**).

**Table 2.** Values of dynamic modulus $E_{vd}$ and respective stiffness of support $k_{Zmin}$.

| The Ground Heterogeneity in a Direction Perpendicular to the Façade | | The Ground Heterogeneity in a Direction Parallel to the Façade | |
|---|---|---|---|
| $E_{vd}$ (**MN/m$^2$**) | $k_Z$ (**kN/m**) | $E_{vd}$ (**MN/m$^2$**) | $k_Z$ (**kN/m**) |
| 2.43 | 35.0 | 2.43 | 35.0 |
| 3.48 | 350.0 | 3.48 | 350.0 |
| 5.22 | 875.0 | 4.64 | 700.0 |
| 8.13 | 1750.0 | 5.80 | 1050.0 |
| 11.03 | 2625.0 | 6.96 | 1400.0 |
| 13.94 | 3500.0 | 8.13 | 1750.0 |
| | | 9.29 | 2100.0 |
| | | 10.45 | 2450.0 |
| | | 11.61 | 2800.0 |
| | | 12.78 | 3150.0 |
| | | 13.94 | 3500.0 |

In all analyses, the scaffolding was loaded with the self-weight and the service load. As recommended by the EN 12811-1 [32], service load was applied to the highest platform and to the platform below it with values equal to 2.0 and 1.0 kN/m$^2$, respectively. The quantities describing the stress of the structure were not taken from calculations directly. The vertical axes of the diagrams in Figures 7–9 show respectively the ratio of maximum value of displacements, axial forces, and normal stresses obtained for the configuration C2 (with the max index) at different values of $\mu$ and for the configuration C3 (with the unmovable index).

If we take into account even small ground subsidence, the displacements increase by approximately 60%. However, if we consider the susceptibility of the subsoil and its heterogeneity, then for changes along the scaffolding and $\mu < 0.2$ a sudden increase in displacement is visible, leading to significant changes in the scaffolding geometry.

The stress in scaffolding elements caused by the ground heterogeneity in the direction parallel to the façade can be reduced using a static scheme in which each plane of frames works separately. This was confirmed by the results for the base jacks in Figure 8. In practice, the uneven subsidence of the ground causes uneven loading of the anchors irrespective of the direction of non-uniformity of the subsoil and significant loading of the bracings, which may become loose during the scaffolding operation. The use of bracings in the scaffolding is to create stiffer sections of the scaffolding, ensuring the rigidity of the structure along the façade. However, at the same time, a part is created in which the two planes of the frames work together and uneven subsidence of the stands along the facade increases the stress in bracings.

The impact of subsoil heterogeneity on the normal stresses in stands, transoms, and girders is smaller (see Figure 9). The change in geometry causes an increase in bending moments and normal stresses in stands, whereas the transoms, especially at ground subsidence in the direction perpendicular to the facade, are additionally bent due to the "shearing" of the frames.

It is also worth noting that in the structure with perfect geometry, but with stiffness compared to the scaffolding in configuration C3, as in Figure 4, the normal stress in the stands was 2.5 times higher while the normal forces were 30% and 35% higher, respectively, in anchors and bracings.

## 4. Conclusions

This paper presents the impact of setting exemplary façade scaffolding on a heterogeneous foundation, an issue that is usually overlooked in scaffolding design. Considerable differences in the foundation of the scaffolding in a perpendicular direction caused an increase in stress in the stands of up to 60%, but also had a negative effect on the transoms. Significant differences in the perpendicular subsidence of scaffolding may occur during the construction of buildings, when the soil near the walls is not compacted. This can also occur during renovation works of existing buildings, if the stands next

to the building are set on a paving and the external stands are set on the ground. For the factor $\mu < 0.25$, the stresses in the stands increased by at least 20% and the forces in the anchors by at least seven times. There were also significant increases in displacements, which is particularly dangerous because the scaffolding may deflect and fall over onto neighboring objects.

In the case of heterogeneity along the scaffolding, the forces in anchors increased up to 10 times. The frames, set on the ground with a lower dynamic deformation module, would be suspended on anchors that are not adapted to carry loads in the vertical direction. A significant increase in normal stress occurred in the stands at $\mu < 0.25$. However, this increase is smaller than for perpendicular heterogeneity. The presented results are theoretical, but this could have occurred, e.g., as a result of ground washout during rainfall.

All scaffolding components and the people that use them are exposed to the negative effects of heterogeneity of the foundation. Therefore, the ground and its compaction should be checked before scaffolding is placed on it. It is necessary to use mudsills that are arranged perpendicular to the facade, and each frame should be set on a separate mudsill. In the future, the permissible value $\mu$ should be determined, and the level of heterogeneity of subsoil parameters should be considered in the design of scaffolding systems and atypical scaffoldings.

Since the results presented in this paper show that the aspect of uneven soil subsidence has a significant impact on the trouble-free use of scaffolding structures, continued research is planned on issues such as: the numerical analysis of stress distribution under one mudsill depending on soil homogeneity and force system transferred from scaffolding; the testing of the sensitivity of scaffolding responses of various sizes and types to foundation parameters; and determination of partial safety factors for scaffolding design that take into account the uncertainty of substrate quality. Additional analyses should be also performed for other types of scaffolding. In the case of modular scaffolding, which has semi-rigid joints between stands and transoms, the static scheme would be different. This is why thorough analyses of the uneven subsidence of the ground are needed for various structures.

**Author Contributions:** Conceptualization, E.B.-B., P.J.-G. and M.P.; methodology, E.B.-B. and M.P.; software, E.B.-B. and P.J.-G.; validation, M.P. and P.J.-G.; investigation, M.P. and P.J.-G.; writing—original draft preparation, E.B.-B.; writing—review and editing, P.J.-G. and M.P.; visualization, P.J.-G. and M.P.; supervision, E.B.-B.; project administration, E.B.-B.; funding acquisition, E.B.-B. All authors have read and agreed to the published version of the manuscript.

**Funding:** This research was funded by the National Centre for Research and Development within the Applied Research Programme, agreement No. PBS3/A2/19/2015 "Modelling of Risk Assessment of Construction Disasters, Accidents, and Dangerous Incidents at Workplaces Using Scaffoldings" and by the Science Financing Subsidy Lublin University of Technology FN16/ILT/2020.

**Acknowledgments:** We would like to thank all colleagues from the Lublin University of Technology, Lodz University of Technology and Wroclaw University of Science and Technology for their engagement in this project. We also thank all companies which agreed to carry out the research at their construction sites for their help.

**Conflicts of Interest:** The authors declare no conflict of interest.

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
