# Peer review of "Influence of Foundation Quality on the Stress in the Elements of Steel Façade Scaffolding"

_buildings, doi:10.3390/buildings10070130_

Round 1

Reviewer 1 Report

The influence of incorrect scaffolding foundation on the effort in its structural members is presented in the paper.

The problem is interesting and important for science and praxis. A good paper and should be published after a minor revision.

Here follows some comments:

Page 6: The use of linear material models allowed to show the differences in normal stresses for each case and configuration. The use of a non-linear model would cause all calculations to end at the material yield strength.

Please, could the authors explain the last sentence. We can generally use materially nonlinear models with adequate strengthening zones behind the plastic plateaus by means of appropriate approximation functions of stress-strain curves.

Author Response

Thank you for your time and thorough review of our submitted manuscript, and for giving us a chance to improve the work. We tried to change the manuscript according to your comments. We hope that the changes done to the paper are in line with your expectations.

Reviewer 2 Report

Dear authors,
your paper is very interesting. In practice, scaffoldings are mostly, and unfortunately, taken for granted, while consequences of their failures can be tremendous. The paper is acceptable, however, here are my comments, questions, and suggestions by which it might be better:
1) it is necessary to determine the type of the scaffold by all categorizations (a.g. bearing capacities, structural design, material, etc.) in the title, in the abstract, and as soon as possible in the text. This also refers to the soil type covered in this research. Please be precise with your findings, because the cannot be general applying to any type of scaffoldings and soil types.
2) it is necessary to provide precise descriptive and analytical parameters of the analyzed scaffolds in the project, and why did you use only one type in this study?
2) in the text it is necessary to explain denotations, also please express loads in kN/m2 instead of Mpa.
3) figure 2 a), b) and c) present a rather poor example of applied safety measures for scaffolding. Reconsider their relevance to the text and the fact that readers might be misguided by them.
4) explain more precisely and thoroughly "The maximum imperfection value measured on the scaffolding L12 was equal to 18.3 cm."
5) explain where and how did you "The first free vibration frequency equal to 1.94 Hz was determined based on scaffolding acceleration measurements."
6) figure 4 has to be more thoroughly explained in the text.
7) conclusions are trivial. Of course that if the foundations and the soil underneath are of low bearing capacity that it will cause additional deformations, as well that the additional bending moments will increase frames normal stress. In conclusion, you have to be very precise with your findings and their interpretation along with their sensitivities and applicabilities, ads well as shortcomings of previous studies in regard to yours.
Kind regards

Author Response

Thank you for your time and thorough review of our submitted manuscript, and for giving us a chance to improve the work. We tried to change the manuscript according to your comments. We hope that the changes done to the paper are in line with your expectations

Reviewer 3 Report

The paper is aimed at numerical investigating the influence of foundation quality on the stress in scaffolding structures and elements. The work tackles an interesting topic and could be suitable for the selected Journal (however a better check should be done by the editors in charge of the manuscript about the suitability of this work in this J).

The work however needs major improvements and clarifications before its approval. In the Reviewer's opinion, the following recommendations/clarifications should be considered for the next submission and I indicate the rejection of the manuscript:

  1. In section 1, after the State of the Art (SoA), the Authors should clarify what are the key novelties of this paper and the main contributions of this work beyond the current SoA. They are missing.
  2. A better description of the model assumptions, type of resolution (FEM?), used element types, boundary conditions, adopted constitutive rules (probably the authors were considered linear elastic materials) etc. should be shown by the Authors.
  3. The work looks like a technical work/report rather than a scientific contribution/paper. It is not clear what is the novelty message beyond the current state of knowledge.
  4. Please add a last paragraph (to be added within the conclusion section) dealing with “future research steps”.

Author Response

(The authors gave the same response as above.)

Round 2

Reviewer 2 Report

Dear authors,
thank you for acknowledging my suggestions. To me your paper seems better in this version, I hope you have the same impression. I'm glad that scaffolding every now and then appears as a research subject, even though it is too rare regarding its scientific and professional potential. In this perspective, your paper is welcomed and I wish you luck in the further publishing steps.
Kind regards

Reviewer 3 Report

The paper can be accepted